# Giant Non-Functioning Pituitary Adenomas: Treatment Considerations

**DOI:** 10.3390/brainsci12091256

**Published:** 2022-09-16

**Authors:** Domenico Solari, Luigi Maria Cavallo, Chiara Graziadio, Sergio Corvino, Ilaria Bove, Felice Esposito, Paolo Cappabianca

**Affiliations:** 1Division of Neurosurgery, Università degli Studi di Napoli “Federico II”, Via Pansini 5, 80131 Naples, Italy; 2Division of Endocrinology, Università degli Studi di Napoli “Federico II”, Via Pansini 5, 80131 Naples, Italy

**Keywords:** pituitary adenomas, endoscopic endonasal surgery, skull base surgery, pituitary/hypothalamus, endocrinology, neurosurgery, giant tumors

## Abstract

Giant pituitary adenomas are a subgroup of pituitary adenomas defined by a diameter greater than 4 cm, and they account for 5–14% of adenomas in surgical series. Because of their growth patterns and locations, often involving critical neurovascular structures, they represent a true surgical challenge, and gross total resection is difficult to achieve. There is no consensus on the optimal surgical strategy for giant pituitary adenomas, and, often, integrated multi-staged treatment strategies have been considered. Transcranial or transsphenoidal approaches, alone or combined, according to tumor and patient features are the two main routes. Each of these strategies has pros and cons. The conventional transcranial approach has for a long time been considered the first choice for the removal of giant pituitary adenomas. Currently, with endoscopic techniques, it is also possible to remove lesions that involve the intradural compartment and the adjacent neurovascular structures with the use of extended approaches. Our policy for the management of these lesions is to adopt the endoscopic endonasal approach as the first choice unless the tumor presents significant intracranial extension that results in it being outside the visibility and maneuverability of the endoscopic endonasal route. In these latter cases, we agree that the transcranial approach is more appropriate. However, accurate preoperative evaluation and refined treatment plans for each patient are mandatory to define a proper strategy in order to achieve the most effective long-term result.

## 1. Introduction

Pituitary adenomas represent the third most common intracranial type of tumors, with a mean prevalence of 16.7% [1,2,3]. The so-called “giant” lesions constitute a subgroup of pituitary adenomas defined by a maximum diameter greater than 4 cm and account for 5–14% of adenomas in surgical series [4,5,6]. According to their functional status, giant pituitary adenomas may be distinguished as secreting or non-functioning, with the latter being the most frequent. Their symptoms of presentation are related to the mass effect, first visual acuity and/or field defects, partial or total hypopituitarism, and sometimes oculomotor and trigeminal first and second branch nerve deficits when cavernous sinus invasion is present. When giant adenomas extend widely in the intracranial compartment, they represent a true surgical challenge because of their size, local invasiveness, and irregular margins, regardless of whether critical neurovascular structure involvement is present. Surgery is the treatment of choice, although complete tumor resection is not commonly carried out; many cases require a multi-modal therapeutic strategy, with adjuvant fractionated stereotactic radiotherapy (SRT) or stereotactic radiosurgery (SRS) being advocated to achieve disease control over a long period of follow-up. The surgical approach to the removal of these lesions depends on two main routes, i.e., the transcranial and transsphenoidal routes, alone or combined, in regard to the anatomical growth pattern and tumor consistency. Each of these strategies has pros and cons and, therefore, a refined treatment plan and a surgical targeted approach are mandatory. Our policy for the management of giant pituitary adenomas is to adopt the endoscopic endonasal extended variation in the technique unless the tumor presents a significant intracranial extension that is outside the visibility and maneuverability of the endoscopic endonasal route [7].

Considering the above, we believe that surgery for giant adenomas needs to be the best possible on the first attempt; nonetheless, it is advisable to tailor the most suitable treatment for each case, accounting for a wide variety of options, e.g., medical, surgical, and radiotherapy, in order to achieve the most effective long-term result [8].

## 2. Clinical Features

Usually, giant adenomas do not present a typical clinical spectrum of signs related to hormone hyperproduction; rather, they cause symptoms related to the mass effect, such as initial visual acuity and/or field defects due to optic nerve/chiasm compression; partial or total hypopituitarism as related to different degrees of pituitary malfunctioning; oculomotor nerve palsy (III-IV-VI) and trigeminal dysesthesia, e.g., for the lesion extent in the parasellar region through the medial walls of the cavernous sinus, resulting in diplopia, ophthalmoplegia, ptosis and/or facial pain [9,10], and, seldom, hydrocephalus.

Patients can also display clinical conditions resulting from the impairment of the normal secretory functions of the anterior pituitary such as fatigue, weakness, and diminished sex drive; patients, especially those over 60, show a particular skin pallor and profound asthenia that causes a progressive reduction in their daily performance and eventual drowsiness.

Frontal or retro orbital headache can occur as a consequence of a tumor stretching the dural layer surrounding the pituitary gland, although it cannot be considered a pathognomonic sign. More rarely, the first clinical manifestation of a giant pituitary adenoma is diabetes insipidus as a result of neurohypophysis and/or pituitary stalk compression: patients complain of irrepressible thirst and polyuria; in such cases, hypernatremia rules out the diagnosis.

## 3. Goals of Surgery

In accordance with the neurological and endocrinological symptomatology, the main surgical goals for patients with giant adenomas should aim to achieve the following [11,12,13,14,15]:Maximal safe tumor removal to grant the relief of mass effect signs;preservation of normal neurologic functions;decompression of the pituitary gland to improve or preserve the residual hormonal function.

However, when dealing with these lesions, it is extremely important to relate the goal of the surgery to the patient’s needs, adopting a suitable strategy from among all the available options of treatment.

## 4. Surgical Techniques

### 4.1. Transsphenoidal Approaches

The transsphenoidal approach represents a safe and minimally invasive route, which relies on the adoption of an anatomical corridor, i.e., the nose, to gain access to the sellar area. It provides direct and good visualization over the pituitary gland and adjacent neurovascular structures. There are two main visualization techniques: the microsurgical and the endoscopic techniques.

#### 4.1.1. Endoscopic Endonasal Approach

This procedure consists of three main aspects: exposure of the lesion, management of the relevant pathology, and reconstruction that, similar to the microsurgical technique, relies on three different phases: the nasal, the sphenoid, and the sellar phases.

The binarial approach is run under visualization of a 0° endoscope of 18 cm in length and 4 mm in diameter; after mucosa decongestion, the middle turbinate is gently dislocated laterally in one nostril, whilst middle turbinectomy on the side where the endoscope is driven along with posterior bilateral ethmoidectomy is recommended. The so-called nasoseptal flap is drawn at the initial stage of the approach according to original description [16,17], whilst the flap is raised and reflected over the skull base defect at the end of the procedure as per our reconstruction policy [18].

The posterior nasal septum is detached and slightly trimmed; the sphenoid anterior wall and the sphenoid septa are removed to achieve a wide exposure of the surgical field.

The endoscopic technique thus provides a panoramic view of the entire sphenoid cavity and allows identification of all the anatomical landmarks (optic and carotid protuberances, clivus, planum sphenoidale, and opto-carotid recess), which is essential for obtaining access to the sellar floor.

The sellar phase follows the same rules as those of the microsurgical transsphenoidal approach.

The opening of the sellar floor depends on the inner anatomical conditions (intact, thinned, or eroded) and should be enlarged as required over to the tuberculum sellae area [19] and planum sphenoidale above and/or the surrounding parasellar areas, as per the paradigm of an extended approach [8,20,21,22]. Hereafter, we describe the crucial steps according to the targeted area.

##### Suprasellar Extension

The opening and the exposure of a giant pituitary adenoma extending in the suprasellar supradiaphragmatic space (Figure 1A–C) [23] requires additional bone removal of the tuberculum sellae and of the sphenoid planum between the protuberances of the optic nerves (Figure 2A).

The endoscopic technique has improved visualization, allowing wider exposition of the suprasellar subchiasmatic area and its vascularization; care must be taken, especially for the small branches of the superior hypophyseal arteries that are often displaced by the tumor (Figure 2B), so as to avoid visual field or pituitary function defects [23]. Again, the adenoma removal has to start from the inferior and lateral angles to achieve lesion debulking and to finally dissect the capsule from the neighboring neurovascular structures. When it is not possible to identify a capsule, tumor removal runs gently with the aid of suction, taking care to not injure the tiny vessels encroached by the lesion itself (Figure 2C); at the end, angled scope (30° or 45°) inspections permit one to explore the anatomical details (Figure 2D) and recognize eventual tumor remnants, which are associated with a higher risk of postoperative intralesional hemorrhage.

##### Parasellar Extension

This approach requires additional bone removal of the carotid protuberance and permits a good exposure of the medial and lateral compartments of the cavernous sinus [24]. In the case of tumors occupying the lateral compartment of cavernous sinus [25], usually displacing the intracavernous carotid artery medially and cranial nerve laterally, the surgical exposure of the lesions requires leveling of the pterygoid process and the bone portion between the vidian canal and foramen rotundum [26]. This technique is helpful for those tumors occupying the entire cavernous sinus, as grade 4 Knosp adenomas, or continuing toward the pterygoid fossa [26,27]. It should be noted that the tumor itself enlarges the C-shaped parasellar segment of the internal carotid artery, thus making the suctioning and the curettage through this corridor easier. Conversely, the approach to the lateral compartment of the cavernous sinus is indicated in the case of tumors involving the entire cavernous sinus. In both cases, the tumor removal proceeds from the extracavernous to the intracavernous portion. In the case of tumors occupying mainly the lateral compartment of the cavernous sinus, the growth of the lesion usually displaces the ICA -Internal Carotid Artery - medially and pushes the cranial nerves laterally. Delicate maneuvers of curettage and suction usually allow the removal of the parasellar portion of the lesion in the same fashion as that for the intrasellar portion [24,28,29].

Furthermore, giant adenomas extending into the nasal or paranasal cavities have to be approached via a lower trajectory. When they extend in the clival area and down to the rhinopharynx, it is important to remove the prow and the floor of the sphenoid sinus. If the whole sphenoid sinus is invaded, the removal of the posterior portion of the nasal septum is mandatory, while to approach a lesion invading the sphenoid sinus’ lateral recess and/or infratemporal fossa, the medial pterygoid processes are removed. In the latter area, care must be taken to avoid damaging the branches of the trigeminal nerve and the extracranial ICA [8,27,28,29].

### 4.2. Reconstruction Technique

After the lesion removal, the closure of the osteodural defect has to be carefully completed to create a watertight barrier and to prevent postoperative CSF – Cerebrospinal Fluid - leakage and related adverse events, including meningitis, brain herniation, and tension pneumocephalus [20,30,31,32,33].

Along with the development of endonasal approaches, various techniques and materials have been adopted in regard to the different requirements, so postoperative CSF leak has deceased from very high rates [17,30,34,35,36,37,38,39,40] down to nearly 5% [38,39,41].

A multi-layer technique, the so-called gasket seal [34] and/or “grandma’s cap” [30], can be used: a heterologous dural substitute graft is placed in the extradural space and wedged with a semisolid buttress. Additionally, the “sandwich technique” or “ bilayer button” [42] made of a few patches of dural substitute or fascia lata stitched with a fat pad is placed in the intradural (fat side) and the extradural compartments and then covered.

A vascularized nasoseptal flap, collected during the procedure, can be used for final reconstruction [16]. Recently, our school defined the “3F technique” [18] as follows: it consists of the use of a fat autologous periumbilical graft fixed with fibrin glue across the intra-extradural as a cork and then covered with a nasoseptal “fresh flap”; the patient is mobilized very early to divert the intracranial pressure and pulsation of the defect (Figure 1D–F).

### 4.3. Transcranial Approaches

The most commonly adopted transcranial approaches are the unilateral subfrontal and the pterional approaches, according to the direction of the lesion growth. The unilateral subfrontal approach is suitable for large suprasellar adenomas with asymmetric lateral extension and with upper prepontine cistern invasion. After bicoronal skin incision and craniotomy, the frontal sinus is opened and packed with temporalis fascia, galea capitis, or a dural substitute. The dura is opened, and the frontal lobe is retracted to visualize and remove the lesion preserving the pituitary stalk and the optic pathways. The pterional or frontolateral approach is used for lesions with an important retrochiasmatic portion as it allows a good view of the inferolateral portion of the frontal lobe and the anterior temporal lobe and proper exposure of the area between the optic nerve, the interior carotid artery, and the third cranial nerve. Craniotomy is carried out around the pterion, the frontal lobe is retracted, the carotid cistern is opened to show the carotid artery, and the tumor is detected in the basal cisterns. When the cavernous sinus is largely involved, Dolenc’s variation may be preferred [43]. Rarely, a bilateral interhemispheric subfrontal approach is used if a wide exposure of the anterior cranial base and of the sellar area is required due to giant adenomas with bilateral suprasellar extension. Nowadays, transcranial approaches are reserved for those tumors showing a significant lateral intracranial extension, i.e., tumors extending in the subfrontal, retrochiasmatic, retrosellar, or temporal areas.

### 4.4. Special Considerations

Currently, transsphenoidal surgery, either with microscopic or endoscopic techniques, is adopted in more than 95% of surgical procedures for the treatment of pituitary adenomas [44,45,46,47]. Nevertheless, here, we would like to underline that there are several conditions, either related to the anatomy route or to the inner features of the lesion, i.e., the size of the sella, the size and the pneumatization of the sphenoid sinus, and/or the carotid arteries’ position and shape [10], which are crucial in determining the surgical strategy.

The initial considerations to be drawn regard the tumor size itself: adenomas with a diameter of 4 cm or greater are considered ‘‘giant’’ [5,38,39,48], but this does not represent an adequate criterion by which to detect the complexity of the surgical removal of these lesions. Hence, lesions growing off the sella, extending upward in the intracranial compartment, compressing the chiasm and/or the third ventricle, eventually breaching the diaphragma, and/or stretching the surrounding neurovascular structures are different as compared to similar-sized lesions extending in the sphenoid sinus, or even down into the nasal cavities.

Thereafter, attention should be focused toward the lateral extent, which could represent a peculiar aspect: giant adenomas mostly present a vertical major axis of growth that reduces the likelihood of neurovascular structure involvement; on the contrary, an eccentric growth into the lateral aspects of the anterior and/or middle cranial fossae with the lesion encroaching the supraclinoid ICA and its branches and/or extending laterally to the optic nerves has to be considered a major issue, notwithstanding the fact that adenomas per se do not invade brain tissue [7,49]. In these latter cases, lesion-related injuries and surgery-related risks are increased as per the results of the mass effect from the compression of perforating vessels of the paraventricular area [50]. As per the general understanding, transcranial surgery should be preferred when tumors present with extensive intracranial invasion into the anterior and/or posterior cranial fossae with or without lateral extension, especially if major vessel involvement is present.

Finally, it is of utmost importance that, regardless of the surgical route adopted, complete removal of intracranial giant adenomas is very difficult [5,38,39,48], and this sheds light upon another major concern of the surgical treatment of these tumors, namely, intralesional hemorrhage of residual tumors. This event might lead to threatening complications, and rarely to death, mostly due to the result of the increase in severe vasospasm phenomena [51,52]. Nonetheless, the occurrence of vasospasm represents a concrete risk even upon complete adenoma removal, and it is strictly related to intense and prolonged manipulation [53,54,55]; its timing and mechanisms mimic those observed in subarachnoid hemorrhage following aneurism rupture. Therefore, it is recommended to adopt measures that help avoid it, such as intraoperative papaverine along with 3H therapy, along with nimodipine in the postoperative course. Close monitoring of the eventual onset of neurological disorders thereafter could facilitate early diagnosis and treatment in accordance with post-SAH vasospasm management guidelines [56].

Considering the above, surgical removal of giant pituitary adenomas requires careful and specific postoperative management and long-term patient follow-up, which can make the difference between a satisfactory and a poor result.

In our school, based upon a long experience with the endoscopic endonasal technique, we began to adopt this approach as a viable strategy for the management of this subset of “intracranial” giant pituitary adenomas. Along with the specific case selection, our decision-making process depends on a detailed preoperative evaluation of the surgical risks as related to the lesion features, especially extensions that are out of the visibility and maneuverability of the endoscopic endonasal route [10]. It should be said that a minority of giant adenomas [57,58] (ranging from 1% to 5%) are prolactin-secreting tumors. In these cases, we prefer, in accordance with our endocrinology team, to have first-line medical treatment starting with low-dose cabergoline; thereafter, dose up-titration follows according to PRL levels. Surgery is carried out based upon the evidences of non-responder tumors or in cases in which massive shrinkage determines CSF leakage.

Pituitary adenomas are a benign and slow-growing disease, but heterogeneous in their peculiar features and aspects; rarely, they present as insidious tumors involving the main neurovascular structures and/or brain tissues, so surgical removal in these cases may be burdened by higher rates of morbidity [9].

For these reasons, we believe that surgery for giant adenomas needs to be the best possible on the first attempt; the treatment strategy of these lesions is a complex battleground where the surgeons along with adjacent specialties should be ready to tailor the most suitable treatment depending on a wide variety of options, e.g., medical, surgical, and radiotherapy, in order to achieve the most effective long-term result [45].

## 5. Treatment Considerations

Giant pituitary adenomas represent a true surgical challenge because of their size; invasiveness into the intracranial compartment; their asymmetric shape; the eventual involvement of the critical neurovascular structures, such as the interpeduncular fossa, the third ventricle, the cavernous sinuses, and the main cranial nerves. Accordingly, the rates of radical resection are equal to or as low as 50% in many published surgical studies, with higher complications when compared to non-giant pituitary adenoma series [5,8,38,39,41,48,59].

There is no consensus on the optimal surgical strategy for giant pituitary adenomas and, often, integrated multi-staged treatment strategies have been considered.

Several authors have attempted to identify the proper management in regard to tumor features. Goel et al. classified these tumors into four grades according to the anatomic extent and the nature of their meningeal coverings [5]: tumors confined within sellar dura and under diaphragma sellae without invasion of the cavernous sinus compartment were classified as grade I; those invading the cavernous sinus through its medial wall were defined as grade II; those that extended into the brain through the dura of the cavernous sinus superior wall are grade III; finally, grade IV tumors are those with supradiaphragmatic-subarachnoid extension. Accordingly, the authors [5] recommended the transsphenoidal approach for grades I-III, and biopsy followed by radiotherapy in the case of grade IV tumors, mostly because of their significant extension and encasement of major blood vessels.

The conventional transcranial approach has been traditionally considered the first choice for the removal of giant pituitary adenomas, above the transsphenoidal approach [8,48,60]. It is effective in removing suprasellar tumors and releasing the optic nerve [43,49,61]. Nevertheless, this approach provides limited visualization of the intrasellar region.

On the other hand, the transsphenoidal approach can be used to achieve complete tumor removal in the case of infradiaphragmatic lesions, despite the presence of a significant tumor suprasellar extension [10,62,63,64,65]. Because of the limitations of this approach, particularly the restricted visualization over the supradiaphragmatic and retrosellar areas, tumor dissection/removal maneuvers of supradiaphragmatic lesions, with or without encasement of the cerebral arteries, can be risky and the degree of resection reduces tremendously [7,8,48,62].

The introduction of the endoscope, adopted over the past two decades for the treatment of pituitary adenomas and other sellar lesions, has improved visualization and allowed the possibility of removing lesions that involve the intradural compartment and the adjacent neurovascular structures [20,46,66,67]. The extended approach allows improved visualization over the whole median skull base and the suprasellar and parasellar regions, ensuring a wider lesion exposure just after the dural opening over the sellar–suprasellar space, thus avoiding any retraction of neurovascular structures. However, there are some conditions, either related to the anatomy of the surgical route or to the inner features of the lesion itself, i.e., the size of the sella, the degree of ossification, the size and the pneumatization of the sphenoid sinus, and/or the carotid arteries’ position and shape, that could render the transsphenoidal approach more troublesome.

It is preferable to start with resection from below rather than from above because the soft consistency itself can allow for an easy removal of the sellar component and, therefore, the resection of the most superior aspects of the tumor; in these terms, a single transsphenoidal approach can be adequate. This policy has been validated by Evans et al. [68], who adopted the endoscopic endonasal approach for all giant adenomas as the initial management. For tumors with significant extension lateral to the optic nerves, they use staged endonasal and craniotomy procedures.

Again, it is worth remembering that partial debulking has been associated with a higher risk of postoperative bleeding, apoplexy, and/or residual suprasellar tumor tissue hemorrhage, resulting in higher rates of morbidity and mortality [69,70] as related to the increased mass effect, compression of the optic pathway, and acute hydrocephalus [49,71,72].

Recently, a combination of transsphenoidal and transcranial approaches has been proposed for the complete resection of giant pituitary adenomas. Some surgeons suggest a two-staged strategy in which one procedure is performed first, followed by the second after weeks or months; other surgeons have used a simultaneous surgical attempt, the so-called “above and below” technique [72,73,74].

The advantages of the latter single-step technique depend on the possibility of achieving a better degree of resection and reducing the postoperative bleeding of the residual tumor; on the other hand, a longer operation time, higher risk of infection, and potential complications associated with both procedures are major drawbacks [71].

From our standpoint, it is worth noting that “size does not matter” [7] in considerations of the main factor defining the optimal surgical treatment for each case of giant pituitary adenoma; rather, the decision-making process should rely on the anatomical relationships, the spread of the growth pattern, and the presumed tumor consistency. When the tumor is firm, rubbery, or fibrous, such as in some cases of recurrent tumors, the possibility of completing safe dissection/removal maneuvers and, therefore, the chance of achieving gross total resection is smaller. On the other hand, a soft consistency and a primarily cystic or hemorrhagic appearance on MR imaging improve the possibilities of successful outcomes in experienced hands, regardless of the approach [49]. Rarely, for these latter features, we adopt the “above and below” technique, with most of the tumor removal maneuvers run via the endonasal corridor and using the transcranial route to check and remove the eventual utmost lateral remnants of the lesions (see Appendix A).

At present, our policy is to reserve transcranial approaches for those tumors showing a significant lateral intracranial extension that results in them being outside the visibility and maneuverability of the endoscopic endonasal route, large and irregular adenomas, and/or tumors extended in the subfrontal, retrochiasmatic, retrosellar, or temporal areas.

## 6. Conclusions

Surgery is the primary treatment for all giant pituitary adenomas, and “maximum-allowed resection” is the goal; nevertheless, no single treatment can be considered effective. There is no standardized approach for these lesions, and the treatment should be tailored to individual cases in regard to the patient and the lesion’s features. Both transsphenoidal and transcranial approaches can be used, alone or flexibly combined, simultaneously or in a two-staged approach.

The endoscopic endonasal route, when adopted in accordance with the possibilities offered by the tumor and the route, provides reasonable resection rates, favorable clinical outcomes with the restoration of vision in approximately 80% of patients, improved hormonal function in 50% of cases, and acceptable complication risks.

For these reasons, the extended endoscopic approach can be considered a valid option in the management of giant pituitary adenomas.

## Figures and Tables

**Figure 1 brainsci-12-01256-f001:**
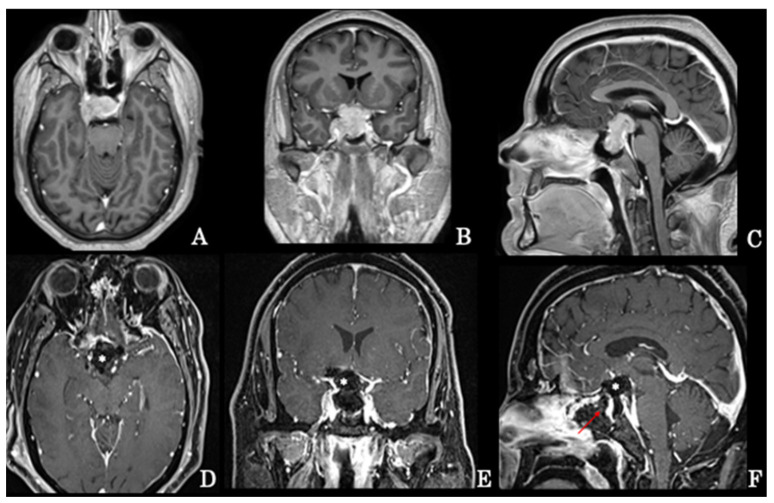
Preoperative axial (**A**), coronal (**B**), and sagittal (**C**) MRI scans of a giant intra-suprasellar pituitary adenoma also extending into the right cavernous sinus that has undergone extended endoscopic removal. Early postoperative axial (**D**), coronal (**E**), and sagittal (**F**) MRI scans revealing the gross total resection of the tumor; the surgical cavity has been filled with autologous fat (*) and the osteodural breach covered with a pedicled nasoseptal flap (arrow) according to the 3F technique.

**Figure 2 brainsci-12-01256-f002:**
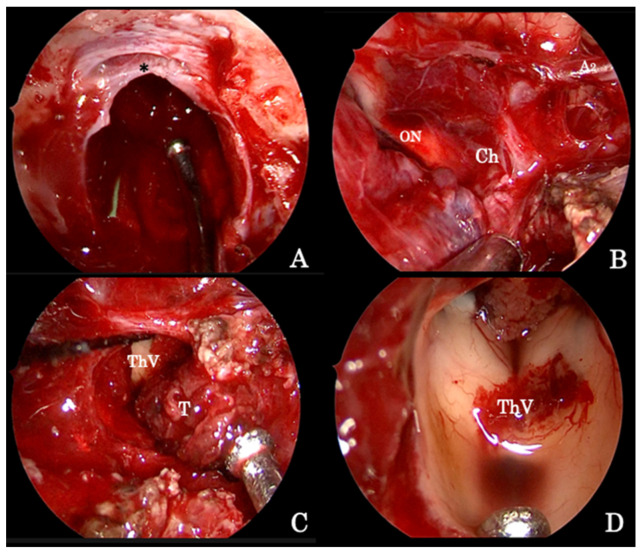
Intraoperative image detailing the endoscopic endonasal procedure for the removal of a giant pituitary adenoma. (**A**) The osteodural breach is created both at the level of the sella and of the sphenoid planum between the protuberances of the optic nerves; the sellar infradiaphragmatic component of the adenoma is removed first. (**B**) Thereafter, the supradiaphragmatic area is exposed and tumor removal is completed by means of standard microsurgical techniques taking care of the dissection lesion off the arachnoid under the close-up endoscopic view. (**C**) The tumor followed along its vertical growth pattern inside the third ventricle cavity, which, at the end, is explored (**D**). A2 segment of the left anterior cerebral artery (A2); optic nerve (ON); optic chiasm (Ch); third ventricle cavity (ThV); tumor (T); diaphragma sellae (*).

## Data Availability

Not applicable.

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
