# Peer review of "Giant Non-Functioning Pituitary Adenomas: Treatment Considerations"

_brainsci, 2022, doi:10.3390/brainsci12091256_

Round 1

Reviewer 1 Report

3. Treatment considerations:

   You wrote that biopsy followed by radiotherapy is recommended in grade 4 tumors. However, thereafter, you also wrote the radical resection with minimum surgical complications should be attempted in giant pituitary adenoma with transsphenoidal and transcranial approach. I think your strategy is umbiguous for grade 4 pituitary adenoma. Also in your institution, in the case of grade 4 pituitary adenoma, would you recommend biopsy and radiation?

Representative case (Fig.1)

   This case is an actually big adenoma, extending laterally and suprasellar part. But not so impresssive. I think you have more impressive case with giant tumor treated with both transsphenoidal and transcranial approach. I would suggest the replace of the representative case if possible.

The strategy of PRL producing pituitady adenoma

 The style of your manusctipt is a review paper. So, I would suggest that you also include the description about the treatment strategy with PRL producing giand pituitary adenoma. Is medical treatment such as cavergorine a first line treatment in your institution? or surgery?   

Author Response

Reviewer’s request: You wrote that biopsy followed by radiotherapy is recommended in grade 4 tumors. However, thereafter, you also wrote the radical resection with minimum surgical complications should be attempted in giant pituitary adenoma with transsphenoidal and transcranial approach. I think your strategy is umbiguous for grade 4 pituitary adenoma. Also in your institution, in the case of grade 4 pituitary adenoma, would you recommend biopsy and radiation?

Authors’ reply: We thank the reviewer for having highlighted this issue. Unfortunately, we did not make it clear; the strategy of biopsy followed by radiotherapy for grade 4 tumor refers to the contribution of Goel (Goel A, Nadkarni T, Muzumdar D, Desai K, Phalke U, Sharma P. Giant pituitary tumors: a study based on surgical treatment of 118 cases. Surg Neurol. 2004;61(5):436-445; discussion 445-436.) and we clarified it. As reported in the text, our policy is to achieve maximal safe resection in all cases, regardless the surgical route adopted.

Reviewer’s request: Representative case (Fig.1): This case is an actually big adenoma, extending laterally and suprasellar part. But not so impresssive. I think you have more impressive case with giant tumor treated with both transsphenoidal and transcranial approach. I would suggest the replace of the representative case if possible.

Authors’ reply: We agree with the reviewer, and therefore we introduced a second illustrative case with video of a giant intracerebral recurrent pituitary adenoma, who has received combined simultaneous endoscopic endonasal and supraorbital approach. Surgical treatment of these lesions should not focus on the size, rather on the peculiar maneuvers required to deal with when entering the supradiaphragmatic subarachnoid space.

Reviewer’s request: The strategy of PRL producing pituitady adenoma. The style of your manusctipt is a review paper. So, I would suggest that you also include the description about the treatment strategy with PRL producing giand pituitary adenoma. Is medical treatment such as cavergorine a first line treatment in your institution? or surgery?   

Authors’ reply: We really appreciated this reviewer’s remark. The present manuscript is intended to provide Federico II School opinion in the field of giant non-functioning adenomas surgical treatment, so we reported mostly useful technical hints. We are aware that a certain percentage of giant adenomas are prolactinomas and there is not a univocal consensus regarding the first line treatment of tumors. Therefore, we introduced several considerations to define our strategy also upon the evidence of a giant prolactinoma.

Reviewer 2 Report

The manuscript "GIANT NON-FUNCTIONING PITUITARY ADENOMAS: TREATMENT CONSIDERATIONS" presents the opinion of the Division of Neurosurgery at the university in Naples on neurosurgical treatment of giant non-functioning pituitary adenomas, making a case for the endoscopic endonasal route. As an opinion paper the argumentation is based on published standards and especially the experience of the authors. No data is presented. (By the way: What does the "Data Availability Statement" mean: "Data is contained within the article"?)

The bibliography is accurate and well used to base the arguments on prior and published research. The manuscript is sub-divided into 6 sections, namely "Introduction", "Clinical features", "Treatment considerations", "Goals of surgery", "Surgical techniques", and "Conclusions". I propose that section 3 "Treatment considerations" be put after section 5 "Surgical techniques" - in my view that would make more sense. Section 5.4 is particularly good to lead from established surgical techniques and their foundation to the nuanced approach as established in Naples, which would then be described and advocated in the treatment considerations section.

There is however one very big downside which is language. Almost every sentence contains errors of spelling, punctuation and/or syntax; the wording is often odd. Please find a native-speaking professional performs some serious copy-editing.

Author Response

Reviewer’s request: The manuscript "GIANT NON-FUNCTIONING PITUITARY ADENOMAS: TREATMENT CONSIDERATIONS" presents the opinion of the Division of Neurosurgery at the university in Naples on neurosurgical treatment of giant non-functioning pituitary adenomas, making a case for the endoscopic endonasal route. As an opinion paper the argumentation is based on published standards and especially the experience of the authors. No data is presented. (By the way: What does the "Data Availability Statement" mean: "Data is contained within the article"?)

Authors’ reply: We thank the reviewer for this comment; we aimed to underlined current policy for the treatment of this tumor and as per request of reviewer 1, we introduced a second illustrative case, with surgical video. So, we clarify the supplementary data statement. 

Reviewer’s request: The bibliography is accurate and well used to base the arguments on prior and published research. The manuscript is sub-divided into 6 sections, namely "Introduction", "Clinical features", "Treatment considerations", "Goals of surgery", "Surgical techniques", and "Conclusions". I propose that section 3 "Treatment considerations" be put after section 5 "Surgical techniques" - in my view that would make more sense. Section 5.4 is particularly good to lead from established surgical techniques and their foundation to the nuanced approach as established in Naples, which would then be described and advocated in the treatment considerations section.

Authors’ reply: We appreciate reviewer suggestions and modified the text accordingly to improve its readability.

Reviewer’s request: There is however one very big downside which is language. Almost every sentence contains errors of spelling, punctuation and/or syntax; the wording is often odd. Please find a native-speaking professional performs some serious copy-editing.

Authors’ reply: We agree with the reviewer, so we submitted to the journal language editing service to improve this aspect.

Round 2

Reviewer 2 Report

The authors have diligently addressed all reviewer comments. By opting to hand over the manuscript to a copyediting service the readability now has improved greatly.